# Evaluation of Strip Meniscometry and Association with Clinical and Demographic Variables in a Community Eye Study (in Bangladesh)

**DOI:** 10.3390/jcm9103366

**Published:** 2020-10-20

**Authors:** Mamunur A.K.M. Rashid, Zhang Zhe Thia, Calesta Hui Yi Teo, Sumaiya Mamun, Hon Shing Ong, Louis Tong

**Affiliations:** 1Ophthalmology, Cornea unit, Al Noor Eye Hospital, 1/9 E, Satmasjid Road, Lalmatia, Dhaka 1207, Bangladesh; mamun3312@gmail.com; 2Yong Loo Lin School of Medicine, National University of Singapore, 10 Medical Drive, Singapore 117597, Singapore; thiazz1995@hotmail.com; 3Ocular Surface Research Group, Singapore Eye Research Institute, The Academia, 20 College Road, Discovery Tower Level 6, Singapore 169856, Singapore; teohuiyicalesta@gmail.com (C.H.Y.T.); honshing@gmail.com (H.S.O.); 4Nutrition and Epidemiology, Institute of Nutrition & Food Science, University of Dhaka, Dhaka 1000, Bangladesh; sumaiya.mamun@du.ac.bd; 5Corneal and External Eye Disease, Singapore National Eye Centre, 11 Third Hospital Ave, Singapore 168751, Singapore; 6Eye-Academic Clinical Program, Duke-NUS Medical School, 8 College Rd, Singapore 169857, Singapore

**Keywords:** dry eye disease, meibomian gland dysfunction, ocular surface disease, dry eye symptoms, questionnaire, Bangladesh, global health, epidemiology

## Abstract

Strip meniscometry (SM) is a relatively new technique for evaluating inferior tear meniscus. We described SM in an epidemiology study and its potential associations with clinical and tear parameters. This cross-sectional study involved 1050 factory garment workers in Gazipur, Bangladesh. The Ocular Surface Disease Index (OSDI) questionnaire and a standard examination for dry eye and meibomian gland dysfunction (MGD), including the five-second SM, were performed by a single ophthalmologist. The participants’ ages were 35.56 ± 12.12 years (range 18–59), with 53.8% women. The overall SM was 7.7 ± 3.6 mm, with skewness of 0.126 and kurtosis of 1.84 in frequency distribution. SM values were significantly lower in men than women, and significantly correlated with schirmers (r = 0.71) and tear break up time (TBUT) (r = 0.89). A lower SM value was associated with higher OSDI, lower Schirmer test, increased MG severity and lower TBUT. In multivariable analysis, when adjusted by age, SM values remained associated with schirmers and TBUT, and inversely associated with OSDI. In a separate regression model, higher SM was associated with increasing age, reduced severity of MGD grading, and increased TBUT. To conclude, SM is a rapid clinical test associated with dry eye symptoms and signs, with findings affected by both tear secretion and tear stability.

## 1. Introduction

Dry Eye Disease (DED), a multifactorial disease of the ocular surface and loss of homeostasis of the tear film, is associated with visual disturbance, symptoms of ocular discomfort, and tear instability [1,2,3]. Tear film and Ocular Surface Society Dry Eye Workshop (TFOS DEWS) II reported that this disease affects about 5–50% of the population. The large variation is due to a large number of research studies on small geographically homogeneous populations. The epidemiology sub-committee emphasized the need to expand prevalence studies to more geographical regions, and to include different races and ethnicities [4]. Dry eye can have a significant impact in patients’ visual function and quality of life, adversely hindering the ability to carry out daily activities, such as reading or driving. This disease of the ocular surface has thus been an increasing public health concern and it poses significant socioeconomic implications [5,6,7,8,9].

Blepharitis and meibomian gland dysfunction (MGD) are major associated factors of DED. MGD is characterized by chronic abnormalities of the meibomian glands, resulting in altered meibum delivery to the tear film, which can result in poor tear film stability or poor breakup times, a type of dry eye classified as evaporative dry eye. On the other hand, dry eye may also be due to aqueous tear deficiency [10,11].

There is scarce data on the epidemiology of dry eye in developing countries within Asia. In developing countries, dry eye has received minimal clinical management and investigative attention compared to other eye diseases. The healthcare burden of dry eye in such countries is essentially unknown [12]. In Indonesia, the age adjusted prevalence of dry eye symptoms is 27.5% (95% CI: 24.8–32.2) [13]. In this report current smoking and pterygium were independent risk factors for the DED [13].

Bangladesh is a country in South Asia located between India and Myanmar, occupying an area of 57,000 square miles, the 8th most populous country in the world. The per capita GDP of Bangladesh is 4992 USD; the country is considered a low income but has the fastest growing real GDP country in the world. As Bangladesh is a populous country of 163 mil, a properly designed epidemiological study will elicit risk factors and knowledge on dry eye that may not be possible in smaller studies elsewhere.

Among the eye diseases in the urban slums of Dhaka, Bangladesh, ocular surface disease forms an important component. A population-based study in Bangladesh found a prevalence of 17.1% for conjunctivitis, 1.4% for blepharitis, 3.2% for DED, and 3.0% for pterygium. This study had limitations in its methodologies of ocular surface diseases assessment. For example, slit-lamp examination was only used in the study for diagnoses and symptoms were not quantified [14]. However, we believe that the reported figures are also underestimated, because the prevalence rates of conditions like dry eye and blepharitis are known to be higher when clinical symptoms are included [14].

Previous studies have shown that strip meniscometry is a rapid clinical test (five seconds per eye) for the assessment of lower meniscal tear volume and may be useful for screening. Unlike tests like Schirmer’s I, it does not induce reflex tearing [15,16]. The results of this test have not been reported in a community setting.

In this study, we performed a community-based study of strip meniscometry in a group of garment factory workers in Bangladesh [17], and potential associations with demographic factors and other clinical factors related to tear function.

## 2. Experimental Section

### 2.1. Study Design

This was a cross-sectional study conducted in a single garment factory in the town of Gazipur, Bangladesh. Participants had given informed verbal consent. The study obtained approval from the local institutional review board (Bangladesh Medical Research Council BMRC/NREC/2017-2018/1157, approved on 2 August 2018), and only utilized clinically accepted procedures and complied with the Tenets of Declaration of Helsinki for human research.

### 2.2. Participants

All participants underwent the following clinical procedures on the initial referral visit. 

### 2.3. Study Procedures

#### 2.3.1. Questionnaire

All participants underwent a symptom evaluation using the Ocular Surface Disease Index (OSDI)© questionnaire (Allergan, Inc., Irvine, CA, USA). Briefly, the OSDI questionnaire consisted of 12 questions, each question graded from 0 to 4. The total OSDI scores on the scale of 0 to 100 were then calculated with the OSDI© (Allergan, Inc., Irvine, CA, USA) formula (sum of scores) × 25/(12 questions), with higher scores representing greater symptoms severity [18].

#### 2.3.2. Strip Meniscometry

Strip Meniscometry (SM) Tube (Echo Electricity Co., Ltd., Fukusima, Japan) has been performed as in previous studies [15,16]. Briefly, one end of the strip was held by the investigator against the lower tear meniscus for five seconds, and the length of wetting of the thread read directly from the millimeter markings provided.

#### 2.3.3. Fluorescein Breakup Time (TBUT)

Briefly, a minimally wet (saline) fluorescein strip (Fluorets, Bausch and Lomb, Rochester, NY, USA) was used to instill fluorescein dye. The procedure for this step has been previously described [19,20].

#### 2.3.4. Schirmer’s I Test

The Schirmer’s I test was done with standard 5 mm wide test strips (Clement Clark^®^, Essex, UK) with a notch for folding, and without prior anesthesia. The strips were positioned over the inferior temporal half of the lower lid margin in both eyes, and participants’ eyes subsequently closed. After 5 min, the extent of tear wetting of the strip was measured to the nearest mm using a ruler [19].

#### 2.3.5. Meibomian Gland Dysfunction Examination

The characteristics of the meibum secreted was evaluated by one ophthalmologist using the right thumb with gentle pressure, under slit-lamp microscopy and graded as follows: 0: clear meibum, 1: colored meibum with normal consistency, 2: viscous meibum, 3: inspissated meibum, and 4: blocked meibomian gland. This was used as a measure of MGD severity.

#### 2.3.6. Slit-Lamp Examination

Other clinical features were examined using a slit lamp biomicroscope. This included scurfing/crusting, subtarsal papillary reaction [19,21], and regularity of the eyelid margin [22]. Corneal sensitivity was also screened using a cotton wisp [23].

### 2.4. Statistical Analysis

Statistical analysis was performed using StataCorp. 2013. Stata Statistical Software: Release 13.1. College Station, TX: StataCorp LP.

The SM variable was evaluated for its frequency distribution and normality. The univariate association of SM with continuous variables was evaluated by categorizing these variables into binary categories, and the association evaluated with the T-test. Whenever there were more than two categories, analysis of variance was used to determine the statistical significance. Univariate logistic regression was performed between meniscometry category and sex, age, ethnicity, and the six ocular surface signs (tear break up time (TBUT), fluorescein corneal staining, Schirmer’s I test, NLMEG, presence of scurf, and inferior fornix papillary grade).

Multivariate logistic regression was performed with meniscometry category as the dependent variable. We performed models using only the clinical signs and demographics of patients, as well as models introducing the predisposing factors of dry eye such as concomitant drugs and medical conditions.

We performed logistic regressions with two thresholds for SM as the dependent variable. Statistical significance was based on alpha of 0.05.

## 3. Results

### 3.1. Clinical and Characteistics of Participants

In this study, mean age of participants was 35.56 ± 12.12 years (range 18 to 59). 53.8% were women. 64.2% (95%CI: 61.2–67.1) of participants had dry eye defined as OSDI > 12. 

### 3.2. Distribution of Strip Meniscometry Readings

The distribution is bimodal and not normal (Figure 1). The mean readings were 7.7 ± 3.6.

### 3.3. Factors Affecting Strip Meniscometry

The factors associated with SM are shown in (Table 1). When performing a T-test to determine the association between MGD and SM, the three grades of MGD were categorized into two categories for the T-test. The smallest two grades of MGD (MGD Types 0 and 1) are combined into one category, whereas MGD Type 2 was used as the other category.

Reduced SM readings were associated with increased OSDI (Figure 2A) (r = −0.72, *p* < 0.001), directly correlated with Schirmer (Figure 2B) (r = 0.71, *p* < 0.001), and associated with increased MG severity (Figure 2C) (*p* < 0.001).

Interestingly, all the 23 subjects with SM < 3 mm had severe dry eye symptoms (OSDI of 33 or more) (*p* < 0.001 on Fischer’s exact probability test).

### 3.4. Multivariate Analysis

We first performed logistic regression with SM categorized as < 7 mm to be low (abnormal), since the mean SM of this study was 7.7 mm. With age, gender, and the clinical parameters as independent variables, we found low values of SM to be significantly associated with higher OSDI and lower Schirmer’s readings (Table 2).

However, when we performed logistic regression with SM categorized as <3 mm to be low (abnormal), we found abnormal SM to be significantly associated with higher OSDI and lower TBUT readings after adjustment for the other variables (Model 4 in Table 3). However there were only 23 or 2.19% of participants with SM less than 3 mm, so it may or may not be possible to uncover all the associated factors with this sample size.

## 4. Discussion

The meniscometry readings in this population of garment factory workers in Bangladesh ranged between 1–16 mm, and significantly lower SM readings were found in males, and in participants with higher OSDI, lower TBUT, lower Schirmer’s readings, and higher MGD grades.

In the first study performed in a clinic in Japan (*n* = 90), significantly lower SM readings were observed among patients with DED. In addition, the Schirmer’s readings, TBUT, mean tear film lipid layer interferometry grade, and vital staining scores were also observed to be lower among patients with DED. With regards to the SM readings, a positive linear correlation exists with the Schirmer’s readings as well as the TBUT [16]. In the second study performed in a clinic in Japan (*n* = 175), significantly lower SM readings were observed among patients with DED, with a positive correlation with graticule scale tear meniscus height (TMH) [15].

There were no previous studies of SM in a community setting or study based on the Bangladeshi participants.

In our study (Table 2), lower Schirmer and higher OSDI were associated with abnormal meniscometry values after adjusting for the other clinical variables. This is not surprising since Schirmer’s I test, apart from a measure of reflex tear secretion, may also be related to the tear volume in the lower meniscus. With a stricter threshold of SM (Table 3), only lower TBUT and higher OSDI were associated with abnormal SM. This may be related to the pre-existing lower meniscal volume to related to tear stability.

This study involved a large sample in the community within a single occupational group. The use of a uniform protocol by a single ophthalmologist provided an accurate and comprehensive analysis and with a higher confidence of estimates. Since the study did not employ a meibomian gland evaluator, the number of meibomian glands yielding liquid meibum was not documented. The number of subjects with SM less than 3 mm was limited. Another possible limitation is that this study involved only one occupational group. We are not certain why the distribution of SM is bimodal.

## 5. Conclusions

In conclusion, SM is inexpensive, quick and easy to perform, and may be a suitable tool for epidemiology studies. Although associated with dry eye symptoms, this test is not equivalent to any one of the conventional tests, but still related to some aspect of tear function. It should be explored in other populations.

## Figures and Tables

**Figure 1 jcm-09-03366-f001:**
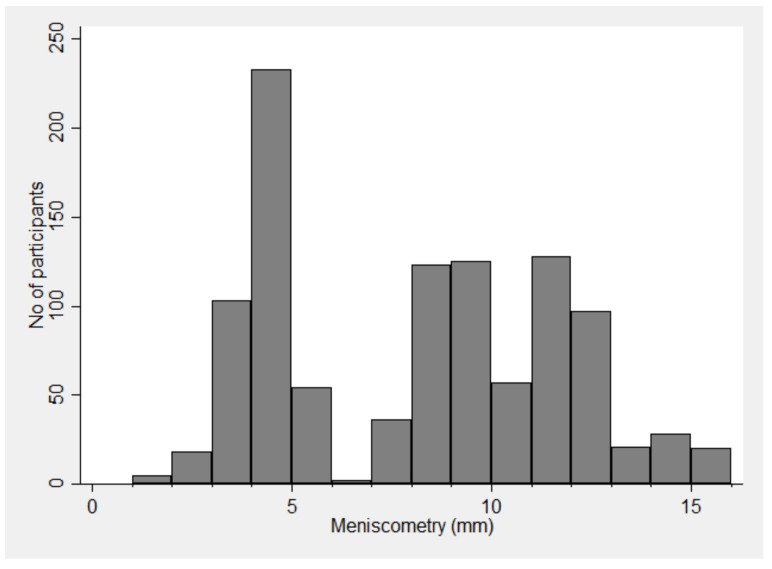
Histogram of strip meniscometry values in factory workers.

**Figure 2 jcm-09-03366-f002:**
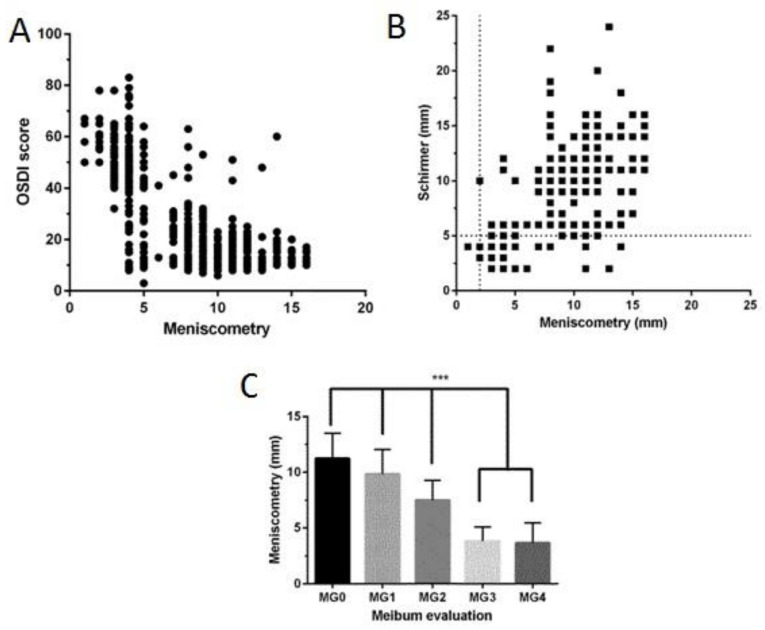
Scatter diagrams showing the relationship between meniscometry and (**A**) Ocular Surface Disease Index (OSDI) scores, (**B**) Schirmer test results. (**C**) Bar graph showing the relationship between meniscometry and meibomian gland dysfunction. MG0: clear meibum, MG1: colored meibum with normal consistency, MG2: viscous meibum, MG3: inspissated meibum, and MG4: blocked meibomian gland. ***: two-tailed p-value for ANOVA and post-hoc tests, *p* < 0.001.

**Table 1 jcm-09-03366-t001:** Summary of strip meniscometry readings in this study.

	Participants N (%)	Strip Meniscometry Reading Mean± Standard Deviation (SD) Median (Min, Max)	*p* Value
Overall	1050 (100%)	7.7 ± 3.6 8 (1, 16)	-
Gender			<0.001
Male	485 (46.19%)	7.1 ± 3.8 7 (1, 16)	
Female	565 (53.81%)	8.1 ± 3.3 9 (2, 16)	
Age			<0.001
<30 years	371 (35.33%)	8.4 ± 3.8 10 (2, 16)	
30–40 years	219 (20.86%)	7.3 ± 3.1 8 (1, 16)	
40–50 years	347 (33.05%)	6.9 ± 3.4 8 (2, 16)	
>50 years	113 (10.76%)	8.1 ± 3.6 9 (2, 14)	
Ocular Surface Disease Index (OSDI)			<0.001
Low (<33)	679 (64.67%)	9.7 ± 2.5 10 (3, 16)	
High (≥33)	371 (35.33%)	3.9 ± 1.4 4 (1, 14)	
Schirmer I test			<0.001
Low (<5 mm)	413 (39.33%)	4.0 ± 1.7 4 (1, 14)	
High (≥5 mm)	637 (60.67%)	10.0 ± 2.3 10 (2, 16)	
Tear Breakup Time (TBUT)			<0.001
Low (<5 s)	369 (35.14%)	3.8 ± 1.3 4 (1, 14)	
High (≥5 s)	681 (64.86%)	9.8 ± 2.5 10 (3, 16)	
Meibomian Gland Dysfunction (MGD) Types			<0.001
Type 0	275 (26.19%)	11.2 ± 2.3 11 (3, 16)	
Type 1	182 (17.33%)	9.8 ± 2.2 10 (4, 16)	
Type 2	593 (56.48%)	5.3 ± 2.4 4 (1, 14)	

**Table 2 jcm-09-03366-t002:** Multiple logistic regression where dependent variable is the abnormal meniscometry values ( < 7 mm).

	Model 1	Model 2 ^†^	Model 3 ^††^	Model 4 ^†††^
Parameters	Unadjusted Odds Ratio (95% Confidence Interval)	Adjusted Odds Ratio (95% Confidence Interval)	Adjusted Odds Ratio (95% Confidence Interval)	Adjusted Odds Ratio (95% Confidence Interval)
Age	1.01 (1.00–1.02) *	1.00 (0.99–1.02)	0.97 (0.95–0.99) *	0.97 (0.94–1.00)
Gender	1.85 (1.44–2.38) *	1.83 (1.42–2.35) *	1.67 (1.06–2.63) *	1.09 (0.61–1.94)
OSDI	1.16 (1.15–1.18) *		1.17 (1.15–1.19) *	1.05 (1.02–1.08) *
TBUT	0.43 (0.39–0.47) *			0.87 (0.72–1.05)
Schirmers	0.17 (0.13–0.23) *			0.43 (0.35–0.54) *
MGD	12.15 (8.96–16.48) *			1.42 (0.79–2.57)

^†^ Adjusted by Age and Gender; ^††^ Adjusted by Age, Gender and OSDI; ^†††^ Adjusted by Age, Gender, ocular surface disease index (OSDI), tear breakup time (TBUT), Schirmers, and meibomian gland dysfunction (MGD); * *p* < 0.05.

**Table 3 jcm-09-03366-t003:** Multiple logistic regression where dependent variable is the abnormal meniscometry values ( < 3 mm).

	Model 1	Model 2 ^†^	Model 3 ^††^	Model 4 ^†††^
Parameters	Unadjusted Odds Ratio (95% Confidence Interval)	Adjusted Odds Ratio (95% Confidence Interval)	Adjusted Odds Ratio (95% Confidence Interval)	Adjusted Odds Ratio (95% Confidence Interval)
Age	1.04 (1.00–1.08)	1.03 (1.00–1.08)	1.02 (0.98–1.07)	1.04 (0.99–1.09)
Gender	2.72 (1.11–6.67) *	2.52 (1.02–6.19) *	1.84 (0.70–4.89)	1.37 (0.49–3.82)
OSDI	1.13 (1.09–1.18) *		1.13 (1.08–1.18) *	1.11 (1.06–1.17) *
TBUT	0.41 (0.29–0.60) *			0.48 (0.30–0.77) *
Schirmers	0.59 (0.44–0.78) *			1.33 (0.94–1.86)
MGD grading	4.80 (2.63–8.77) *			1.67 (0.60–4.68)

^†^ Adjusted by Age and Gender; ^††^ Adjusted by Age, Gender and OSDI; ^†††^Adjusted by Age, Gender, ocular surface disease index (OSDI), tear breakup time (TBUT), Schirmers, and meibomian gland dysfunction (MGD); * *p* < 0.05.

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
