# Peer review of "Evaluation of Strip Meniscometry and Association with Clinical and Demographic Variables in a Community Eye Study (in Bangladesh)"

_jcm, 2020, doi:10.3390/jcm9103366_

Round 1
Reviewer 1 Report
Thank you for allowing me to review your paper. I have a few minor comments for your consideration.
- Line 29 to 32: SM needs to be more clear. How is the SM associated with increased age, etc? It is the value of the SM? High or low?
- Line 61: change to Bangladesh is $4992 USD; the...
- Line 66 to 72: A population-based study is discussed, sharing 3.2% prevalence of DED. It is also stated that these prevalence rates are high. Yet on line 41 it states DED affects about 5-50% of the population. Consider removing the sentence stating "Nevertheless, these prevalence rates are high."
- Line 91: ...(OSDI) (c) add 'questionnaire' after the (c).
- Line 96: Kindly share the strip details. Who manufactures it, etc. similar to the Schirmer's I test details seen on line 103.
- Line 100: Include the manufacture of the fluorescein strips used.
- Line 109: how was the meibum secreted? Kindly include in this section.
- Line 136: 64.2% (95% CI: 61.2-67.1) of participants...
- Line 156:less than 3 mm, so it may or may not be possible...
- Discussion section: It would be nice to see the study limitations discussed too. One might point out the limited number of subjects with SM less than 3 mm (2.19%). Another possible limitation is that this is one occupational group.
- Global: need to identify what the * symbol means for all tables in the footnote under the table.
- Global: All (C) symbols should be of the same size and superscript. Line 91 and line 93 the (c) symbol is different.
Author Response
- We inserted the word 'higher SM' was associated with...
- Changed
- We deleted that sentence as suggested.
- We added the word 'questionnaire' after the (c)
- In Line 96 we inserted: SM Tube (Echo Electricity Co.,Ltd., Fukusima, Japan)
- We inserted this: (Fluorets, Bausch and Lomb, Rochester, NY)
- We added: ...by one ophthalmologist using the right thumb with gentle pressure, under slit lamp microscopy...
- We added ...of...
- We added... not...
- We added to the discussion: "The number of subjects with SM less than 3 mm was limited. Another possible limitation is that this study involved only one occupational group."
- We added * p<0.05 to the footnotes of Tables 2 and 3.
- We amended the copyright symbol on line 91 to be superscript too.
Reviewer 2 Report
The authors analyzed tear secretion test results in garment factory workers by several different methods including strip meniscometry (SM). I acknowledge the significance and uniqueness of the authors’ work conducted in their country, however, considering the concerns below including ethical issues, the contents of the current manuscript have not yet reached a clinical significance for global readers who are in contact with patients with dry eye diseases. Above all, it is not clear the authors recommend SM to what kind of population.
- The authors stated the significance of the study conducted in their country in Introduction section. However, it is not clear whether the working population of 1,050 people in a garment factory reflect the entire population of the country, even though the sample size is relatively larger than preceding studies.
- The Experimental section needs more details. What were inclusion/exclusion criteria for the recruitment for the participants? There is an ethical concern arisen from lack of written informed consent. The permission number awarded by the ethical committee should be revealed.
- Correlation analysis results, p values and so on, for Figure 2 (between meniscometry and OSDI score and between meniscometry and Schirmer, are they Fig. 2A and 2B respectively?) should be described.
- The Discussion section is too short for the results of the study. There is a lack of discussion on the findings the bimodal pattern of meniscometry in Figure 1. Also, there is a lack of discussion on the findings in Table 1. In Table 1, decreased SM results were more prominent in male than in female, which is an interesting data because Sjogren’s syndrome, a representative autoimmune disease with dry eye, is common among female.
- Please explain what asterisks mean in Tables.
Author Response
- However, it is not clear whether the working population of 1,050 people in a garment factory reflect the entire population of the country, even though the sample size is relatively larger than preceding studies.
We added to the discussion: "Another possible limitation is that this study involved only one occupational group."
- The Experimental section needs more details. What were inclusion/exclusion criteria for the recruitment for the participants? There is an ethical concern arisen from lack of written informed consent. The permission number awarded by the ethical committee should be revealed.
We did not use any inclusion or exclusion criteria because we aimed for 100% of the study population to be included, as long as verbal consent was given.
We added: (Bangladesh Medical Research Council BMRC/NREC/2017-2018/1157, approved on 2 August 2018)
- Correlation analysis results, p values and so on, for Figure 2 (between meniscometry and OSDI score and between meniscometry and Schirmer, are they Fig. 2A and 2B respectively?) should be described.
We have labelled the figure with panels A, B and C and added the r and p values into the results section. The revised sentence now reads:
Reduced SM readings were associated with increased OSDI (Figure 2A)(r=-0.72, p<0.001), directly correlated with Schirmer (Figure 2B) (r=0.71, p<0.001), and associated with increased MG severity (Figure 2C) (p<0.001).
We also clarified the Fig 2 legend with:
"Scatter diagrams showing relationship between meniscometry and A. OSDI scores, B. Schirmer test results. C. Bar graph showoing the relationship between meniscometry and meibomian gland dysfunction." and "*** p<0.001"
- The Discussion section is too short for the results of the study. There is a lack of discussion on the findings the bimodal pattern of meniscometry in Figure 1. Also, there is a lack of discussion on the findings in Table 1. In Table 1, decreased SM results were more prominent in male than in female, which is an interesting data because Sjogren’s syndrome, a representative autoimmune disease with dry eye, is common among female.
This study did not involve any participants with Sjogren's syndrome. We added these to the discussion:
"The number of subjects with SM less than 3 mm was limited. Another possible limitation is that this study involved only one occupational group. We are not certain why the distribution of SM is bimodal."
- Please explain what asterisks mean in Tables
We added the explanation in the footnotes of Tables 2 and 3.
Reviewer 3 Report
no additional comments.
Author Response
We have improved the description of the introduction, methods, results, and discussion through the changes suggested by reviewers 1 and 2.
Thank you.
Reviewer 4 Report
The topic of this manuscript was helpful for clinicians, and it was well-written. The analysis of the results were appropriate.
Round 2
Reviewer 2 Report
(None)